# Putting An End to End-to-End:
# Gradient-Isolated Learning of Representations

Sindy Löwe*      Peter O'Connor      Bastiaan S. Veeling*

AMLab
University of Amsterdam
loewe.sindy@gmail.com, basveeling@gmail.com

## Abstract

We propose a novel deep learning method for local self-supervised representation learning that does not require labels nor end-to-end backpropagation but exploits the natural order in data instead. Inspired by the observation that biological neural networks appear to learn without backpropagating a global error signal, we split a deep neural network into a stack of gradient-isolated modules. Each module is trained to maximally preserve the information of its inputs using the InfoNCE bound from Oord et al. [2018]. Despite this greedy training, we demonstrate that each module improves upon the output of its predecessor, and that the representations created by the top module yield highly competitive results on downstream classification tasks in the audio and visual domain. The proposal enables optimizing modules asynchronously, allowing large-scale distributed training of very deep neural networks on unlabelled datasets.

## 1   Introduction

Modern deep learning models are typically optimized using end-to-end backpropagation and a global, supervised loss function. Although empirically proven to be highly successful [Krizhevsky et al., 2012, Szegedy et al., 2015], this approach is considered biologically implausible. For one, supervised learning requires large labeled datasets to ensure generalization. In contrast, children can learn to recognize a new category based on a handful of samples. Additionally, despite some evidence for top-down connections in the brain, there does not appear to be a global objective that is optimized by backpropagating error signals [Crick, 1989, Marblestone et al., 2016]. Instead, the biological brain is highly modular and learns predominantly based on local information [Caporale and Dan, 2008].

In addition to lacking a natural counterpart, the supervised training of neural networks with end-to-end backpropagation suffers from practical disadvantages as well. Supervised learning requires labeled inputs, which are expensive to obtain. As a result, it is not applicable to the majority of available data, and suffers from a higher risk of overfitting, as the number of parameters required for a deep model often exceeds the number of labeled datapoints at hand. At the same time, end-to-end backpropagation creates a substantial memory overhead in a naïve implementation, as the entire computational graph, including all parameters, activations and gradients, needs to fit in a processing unit's working memory. Current approaches to prevent this require either the recomputation of intermediate outputs [Salimans and Bulatov, 2017] or expensive reversible layers [Jacobsen et al., 2018]. This inhibits the application of deep learning models to high-dimensional input data that surpass current memory constraints. This problem is perpetuated as end-to-end training does not allow for an exact way of asynchronously optimizing individual layers [Jaderberg et al., 2017]. In a globally optimized network, every layer needs to wait for its predecessors to provide its inputs, as well as for its successors to provide gradients.

---

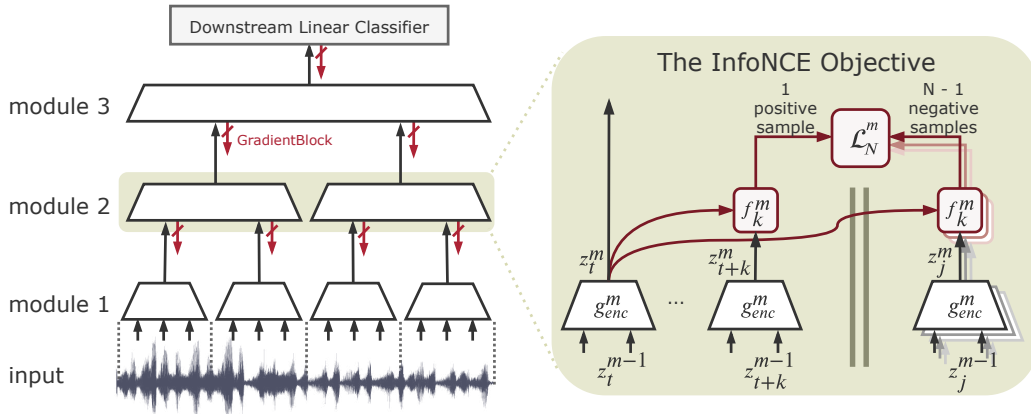

**Figure 1:** The Greedy InfoMax Learning Approach. **(Left)** For the self-supervised learning of representations, we stack a number of modules through which the input is forward-propagated in the usual way, but gradients do not propagate backward. Instead, every module is trained greedily using a local loss. **(Right)** Every encoding module maps its inputs $z_t^{m-1}$ at time-step $t$ to $g_{enc}^m(\text{GradientBlock}(z_t^{m-1})) = z_t^m$, which is used as the input for the following module. The InfoNCE objective is used for its greedy optimization. This loss is calculated by contrasting the predictions of a module for its future representations $z_{t+k}^m$ against negative samples $z_j^m$, which enforces each module to maximally preserve the information of its inputs. We optionally employ an additional autoregressive module $g_{ar}$, which is not depicted here.

This forward and backward locking of the network caused by the backpropagation algorithm impedes the efficiency of hardware accelerator design due to a lack of locality.

In this paper, we introduce a novel learning approach, *Greedy InfoMax* (GIM), that improves upon these problems. Drawing inspiration from biological constraints, we remove end-to-end backpropagation by dividing a deep architecture into gradient-isolated modules that we train using a greedy, self-supervised loss per module. Given unlabeled high-dimensional sequential or spatial data, we encode it iteratively, module by module. By using a loss that enforces the individual modules to maximally preserve the information of their inputs, we enable the stacked model to collectively create compact representations that can be used for downstream tasks. Our contributions are as follows:[1]

- The proposed Greedy InfoMax algorithm achieves strong performance on audio and image classification tasks despite greedy self-supervised training.

- This enables asynchronous, decoupled training of neural networks, allowing for training arbitrarily deep networks on larger-than-memory input data.

- We show that mutual information maximization is especially suited for layer-by-layer greedy optimization, and argue that this reduces the problem of vanishing gradients.

## 2   Background

In order to create compact representations from data that are useful for downstream tasks, we assume that natural data exhibit so-called *slow features* [Wiskott and Sejnowski, 2002]. It is theorized that such features are highly effective for downstream tasks such as object detection or speech recognition. To illustrate: a patch of a few milliseconds of raw speech utterances shares information with neighboring patches such as the speaker identity, emotion, and phonemes, while it does not necessarily share these with random patches drawn from other utterances. Similarly, a small patch from a natural image shares many aspects with neighboring patches such as the depicted object or lighting conditions.

Recent work [Hjelm et al., 2019, Oord et al., 2018] has proposed how we can exploit this to learn representations that maximize the *mutual information* shared among neighbors. In this work, we focus specifically on Contrastive Predictive Coding (CPC) [Oord et al., 2018]. This self-supervised

end-to-end learning approach extracts useful representations from sequential inputs by maximizing the mutual information between the extracted representations of temporally nearby patches.

In order to achieve this, CPC first processes the sequential input signal $x$ using a deep encoding model $g_{enc}(x_t) = z_t$, and additionally produces a representation $c_t$ that aggregates the information of all patches up to time-step $t$ using an autoregressive model $g_{ar}(z_{0:t}) = c_t$. Then, the mutual information between the extracted representations $z_{t+k}$ and $c_t$ of temporally nearby patches is maximized by employing a specifically designed global probabilistic loss: Following the principles of Noise Contrastive Estimation (NCE) [Gutmann and Hyvärinen, 2010], CPC takes a bag $X = \{z_{t+k}, z_{j_1}, z_{j_2}, ...z_{j_{N-1}}\}$ for each delay $k$, with one "positive sample" $z_{t+k}$ which is the encoding of the input that follows $k$ time-steps after $c_t$, and $N - 1$ "negative samples" $z_{j_n}$ which are uniformly drawn from all available encoded input sequences.

Each pair of encodings $(z_j, c_t)$ is scored using a function $f(\cdot)$ to predict how likely it is that the given $z_j$ is the positive sample $z_{t+k}$. In practice, Oord et al. [2018] use a log-bilinear model $f_k(z_j, c_t) = \exp\left(z_j^T W_k c_t\right)$ with a unique weight-matrix $W_k$ for each $k$-steps-ahead prediction. The scores from $f(\cdot)$ are used to predict which sample in the bag $X$ is correct, leading to the InfoNCE loss:

$$\mathcal{L}_N = -\sum_k \mathbb{E}_X \left[ \log \frac{f_k(z_{t+k}, c_t)}{\sum_{z_j \in X} f_k(z_j, c_t)} \right]. \tag{1}$$

This loss is used to optimize both the encoding model $g_{enc}$ and the auto-regressive model $g_{ar}$ to extract the features that are consistent over neighboring patches but which diverge between random pairs of patches. At the same time, the scoring model $f_k$ learns to use those features to correctly classify the matching pair. In practice, the loss is trained using stochastic gradient descent with mini-batches drawn from a large dataset of sequences, and negative samples drawn uniformly from all sequences in the minibatch. Note, that no min-max issues arise as found in adversarial training.

As a result of this configuration, one can derive that the optimal solution for $f$ is proportional to the following density ratio [Oord et al., 2018]:

$$f_k(z_{t+k}, c_t) \propto \frac{p(z_{t+k}|c_t)}{p(z_{t+k})}. \tag{2}$$

This insight allows us to reformulate $-\mathcal{L}_N$ as a lower bound on the mutual information $I(z_{t+k}, c_t)$, as demonstrated in the appendix of Oord et al. [2018] and proven by Poole et al. [2018]. Minimizing the loss $\mathcal{L}_N$ thus optimizes the mutual information between consecutive patch representations $I(z_{t+k}, c_t)$, which in itself lower bounds the mutual information $I(x_{t+k}, c_t)$ between the future input $x_{t+k}$ and the current representation $c_t$. Hyvarinen and Morioka [2016] show that a similar patch-contrastive setup leads to the extraction of a set of conditionally-independent components, such as Gabor-like filters found in the early biological vision system.

**Layer-wise Information Preservation in Neuroscience**   Linsker [1988] developed the InfoMax principle in 1988. It theorizes that the brain learns to process its perceptions by maximally preserving the information of the input activities in each layer. On top of this, neuroscience suggests that the brain predicts its future inputs and learns by minimizing this prediction error [Friston, 2010]. Empirical evidence indicates, for example, that retinal cells carry significant mutual information between the current and the future state of their own activity [Palmer et al., 2015]. Rao and Ballard [1999] indicate that this process may happen at each layer within the brain. Our proposal draws motivation from these theories, resulting in a method that learns to preserve the information between the input and the output of each layer by learning representations that are predictive of future inputs.

## 3   Greedy InfoMax

In this paper, we pose the question if we can effectively optimize the mutual information between representations at each layer of a model in isolation, enjoying the many practical benefits that greedy training (decoupled, isolated training of parts of a model) provides. In doing so, we introduce a novel approach for self-supervised representation learning: Greedy InfoMax (GIM). As depicted on the left side of Figure 1, we take a conventional deep learning architecture and divide it by depth into a stack of $M$ modules. This decoupling can happen at the individual layer level or, for example, at the level

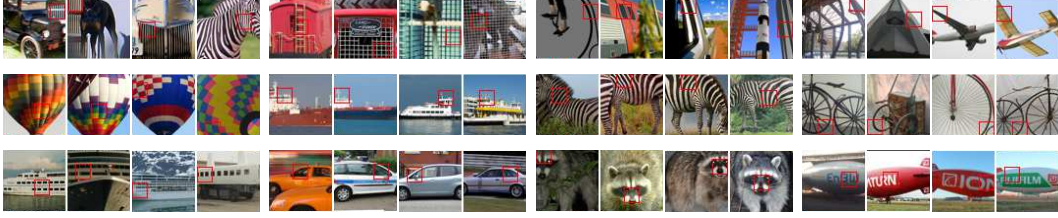

**Figure 2:** Groups of 4 image patches that excite a specific neuron, at *3* levels in the model (**rows**). Despite unsupervised greedy training, neurons appear to extract increasingly semantic features. Best viewed on screen.

of blocks found in residual networks [He et al., 2016]. Rather than training this model end-to-end, we prevent gradients from flowing between modules and employ a local self-supervised loss instead, additionally reducing the issue of vanishing gradients.

As shown on the right side of Figure 1, each encoding module $g_{enc}^m$ within our architecture maps the output from the previous module $z_t^{m-1}$ to an encoding $z_t^m = g_{enc}^m(\text{GradientBlock}(z_t^{m-1}))$. No gradients are flowing between modules, which is enforced using a gradient blocking operator defined as $\text{GradientBlock}(x) \triangleq x, \nabla \text{GradientBlock}(x) \triangleq 0$. Oord et al. [2018] propose to use the output of an autoregressive model $g_{ar}(z_{0:t}) = c_t$ to contrast against future predictions $z_{t+k}$. However, our preliminary results showed that this did not improve results if applied at every module in the stack and optimizing it requires backpropagation through time, which is considered biologically implausible. Therefore, we train each module $g_{enc}^m$ using the following module-local InfoNCE loss:

$$f_k^m(z_{t+k}^m, z_t^m) = \exp\left(z_{t+k}^m{}^T W_k^m z_t^m\right) \tag{3}$$

$$\mathcal{L}_N^m = -\sum_k \mathbb{E}_X \left[ \log \frac{f_k^m(z_{t+k}^m, z_t^m)}{\sum_{z_j^m \in X} f_k^m(z_j^m, z_t^m)} \right]. \tag{4}$$

After convergence of all modules, the scoring functions $f_k^m(\cdot)$ can be discarded, leaving a conventional feed-forward neural network architecture that extracts features $z_t^M$ for downstream tasks:

$$z_t^M = g_{enc}^M\left(g_{enc}^{M-1}\left(\cdots g_{enc}^1\left(x_t\right)\right)\right). \tag{5}$$

For certain downstream tasks, a broad context is essential. For example, in speech recognition, the receptive field of $z_t^M$ might not carry enough information to distinguish phonetic structures. To provide this context, we reintroduce the autoregressive model $g_{ar}$ as an independent module that we optionally append to the stack of encoding modules, resulting in a context-aggregate representation $c_t^M = g_{ar}^M\left(\text{GradientBlock}\left(z_{0:t}^{M-1}\right)\right)$. In practice, a GRU or PixelCNN-style model can serve in this role. We train this module independently using the following altered scoring function:

$$f_k^M(z_{t+k}^{M-1}, c_t^M) = \exp\left(\text{GradientBlock}\left(z_{t+k}^{M-1}\right)^T W_k^M c_t^M\right). \tag{6}$$

**Iterative Mutual Information Maximization**   Similarly to the InfoNCE loss in Equation (1), our module-local InfoNCE loss in Equation (4) maximizes a lower bound on the mutual information $I(z_{t+k}^m, z_t^m)$ between nearby patch representations, encouraging the extraction of slow features.

Most importantly, it follows from Oord et al. [2018], that the module-local InfoNCE loss also maximizes the lower bound of the mutual information $I(z_{t+k}^{m-1}, z_t^m)$ between the future input to a module and its current representation. This can be seen as a maximization of the mutual information between the input and the output of a module, subject to the constraint of temporal disparity. Thus, the InfoNCE loss can successfully enforce each module to maximally preserve the information of its inputs, while providing the necessary regularization [Hu et al., 2017, Krause et al., 2010] for circumventing degenerate solutions. These factors contribute to ensuring that the greedily optimized modules provide meaningful inputs to their successors and that the network as a whole provides useful features for downstream tasks without the use of a global error signal.

**Practical Benefits**   Applying GIM to high-dimensional inputs, we can optimize each module in sequence to decrease the memory costs during training. In the most memory-constrained scenario,

**Table 1:** STL-10 classification results on the test set. The GIM model outperforms the CPC model, despite a lack of end-to-end backpropagation and without the use of a global objective. ($\pm$ standard deviation over 4 training runs.)

| Method | Accuracy (%) |
|---|---|
| Deep InfoMax [Hjelm et al., 2019] | 78.2 |
| Predsim [Nøkland and Eidnes, 2019] | 80.8 |
| Randomly initialized | 27.0 |
| Supervised | 71.4 |
| Greedy Supervised | 65.2 |
| CPC | $80.5 \pm 3.1$ |
| **Greedy InfoMax (GIM)** | $\mathbf{81.9 \pm 0.3}$ |

**Table 2:** GPU memory consumption during training. All models consist of the ResNet-50 architecture and only differ in their training approach. GIM allows efficient greedy training.

| Method | GPU memory (GB) |
|---|---|
| Supervised | 6.3 |
| CPC | 7.7 |
| GIM - all modules | 7.0 |
| GIM - 1st module | **2.5** |

individual modules can be trained, frozen, and their outputs stored as a dataset for the next module, which effectively removes the depth of the network as a factor of the memory complexity.

Additionally, GIM allows for training models on larger-than-memory input data with architectures that would otherwise exceed memory limitations. Leveraging the conventional pooling and strided layers found in common network architectures, we can start with small patches of the input, greedily train the first module, extract the now compressed representation spanning larger windows of the input and train the following module using these.

Last but not least, GIM provides a highly flexible framework for the training of neural networks. It enables the training of individual parts of an architecture at varying update frequencies. When a higher level of abstraction is needed, GIM allows for adding new modules on top at any moment of the optimization process without having to fine-tune previous results.

## 4 Experiments

We test the applicability of the GIM approach to the visual and audio domain. In both settings, a feature-extraction model is divided by depth into modules and trained without labels using GIM. The representations created by the final (frozen) module are then used as the input for a linear classifier, whose accuracy scores provide us with a proxy for the quality and generalizability of the representations created by the self-supervised model.

### 4.1 Vision

To apply Greedy InfoMax to natural images, we impose a top-down ordering on 2D images. We follow Hénaff et al. [2019], Oord et al. [2018] by extracting a grid of partly-overlapping patches from the image to restrict the receptive fields of the representations. For each patch $x_{i,j}$ in row $i$ and column $j$ of this grid, we predict up to K patches $x_{i+K,j}$ in the rows underneath, skipping the first overlapping patch $x_{i+1,j}$. Random contrastive samples are drawn with replacement from all samples available inside a batch, using 16 contrastive samples for each evaluation of the loss. No autoregressive module $g_{ar}$ is used for GIM in this regime.

**Experimental Details**  We focus on the STL-10 dataset [Coates et al., 2011] which provides an additional unlabeled training dataset. For data augmentation, we take random $64 \times 64$ crops from the $96 \times 96$ images, flip horizontally with probability $0.5$ and convert to grayscale. We divide each image of $64 \times 64$ pixels into a total of $7 \times 7$ local patches, each of size $16 \times 16$ with 8 pixels overlap. The patches are encoded by a ResNet-50 v2 model [He et al., 2016] without batch normalization [Ioffe and Szegedy, 2015]. We split the model into three gradient-isolated modules that we train in sync and with a constant learning rate. After convergence, a linear classifier is trained – without finetuning the representations – using a conventional softmax activation and cross-entropy loss. This linear classifier accepts the patch representations $z_{i,j}^M$ from the final module and first average-pools these, resulting in a single vector representation $z^M$. Remaining implementation details are presented in Appendix A.1.

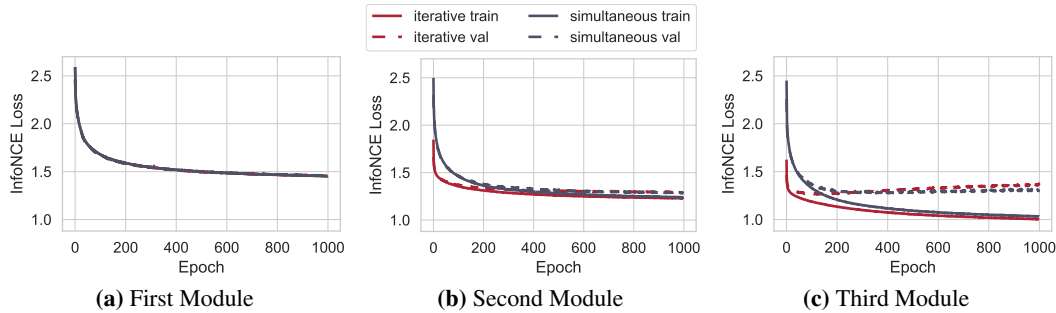

**Figure 3:** Training curves for optimizing all modules *simultaneously* (blue) or *iteratively*, one at a time (red). While there is no difference in the training methods for the first module (**a**), later modules (**b, c**) start out with a lower loss and tend to overfit more when trained iteratively on top of already converged modules.

**Results** As shown in Table 1, *Greedy InfoMax (GIM)* outperforms its end-to-end trained *CPC* counterpart, despite its unsupervised features being optimized greedily without any backpropagation between modules. An equivalent *randomly initialized* feature extraction model exhibits poor performance, showing that GIM extracts useful features. Training the feature extraction model end-to-end and fully *supervised* performs worse, likely due to the small size of the annotated dataset resulting in overfitting. Although this could potentially be circumvented through regularization techniques [De-Vries and Taylor, 2017], the self-supervised methods do not appear to require regularization as they benefit from the full unlabeled dataset. Using a *greedy supervised* approach for training the feature model impedes performance, which suggests that mutual information maximization is unique in its direct applicability to greedy optimization.

In comparison with the recently proposed *Deep InfoMax* model from Hjelm et al. [2019] which uses a slightly different end-to-end mutual information maximization approach, AlexNet [Krizhevsky et al., 2012] as their feature-extraction model and an additional hidden layer in the supervised classification model, GIM comes out favorably. Finally, we see that we outperform the state-of-the-art biologically inspired *Predsim* model from Nøkland and Eidnes [2019], which trains individual layers of a VGG like architecture [Simonyan and Zisserman, 2014] using two supervised loss functions.

In Figure 2, we visualize patches that neurons in intermediate modules of the GIM model are sensitive to. This demonstrates that modules later in the model focus on increasingly abstract features. Overall, the results demonstrate that complicated visual tasks can be approached using greedy self-supervised optimization, which can utilize large-scale unlabeled datasets.

**Asynchronous memory usage** GIM provides a significant practical advantage arising from the greedy nature of optimization: modules can be trained in isolation given cached outputs from previous modules, effectively removing the depth of the network as a factor of the memory complexity. Measuring the allocated GPU memory of the previously studied models during training (Table 2), indicates that this theoretical benefit holds in practice as well. After splitting the architecture into three separately trainable modules, we can reduce the GPU memory consumption by a factor of 2.8 by training the modules asynchronously (*GIM - 1st module*) compared to training them simultaneously (*GIM - all modules*).

We evaluate whether training modules asynchronously influences the quality of the representations. Focusing on the extreme case, we optimize each module until convergence and fix its parameters, before we train the next module on top of it. This *iteratively* trained model achieves an accuracy of 79.8% on the image classification downstream task. Thus, the performance declines slightly in comparison to the *simultaneously* trained model, as previously shown in Table 1 with 81.9% accuracy.

The training curves of the two models as shown in Figure 3 provide some insight into this decreased performance. The learning curves of the first module (Figure 3a) reflect that there is no difference in its training in the two models. Modules two and three (Figures 3b and 3c), however, reveal a crucial difference. The *iteratively* trained modules show a larger divergence between the training and validation loss, indicating stronger overfitting. We tentatively attribute this to the regularizing effect from the initially noisy inputs received by the higher modules when training simultaneously.

**Table 3:** Results for classifying speaker identity and phone labels in the LibriSpeech dataset. All models use the same audio input sizes and the same architecture. Greedy InfoMax creates representations that are useful for audio classification tasks despite its greedy training and lack of a global objective.

| Method | Phone Classification Accuracy (%) | Speaker Classification Accuracy (%) |
|---|---|---|
| Randomly initialized [b] | 27.6 | 1.9 |
| MFCC features [b] | 39.7 | 17.6 |
| Supervised | 77.7 | 98.9 |
| Greedy Supervised | 73.4 | 98.7 |
| CPC [Oord et al., 2018] [a] | 64.9 | 99.6 |
| Greedy InfoMax (GIM) | 62.5 | 99.4 |

[a]In the original implementation, Oord et al. [2018] achieved $64.6\%$ for the phone and $97.4\%$ for the speaker classification task. [b]Baseline results from Oord et al. [2018].

## 4.2 Audio

We evaluate GIM in the audio domain on the sequence-global task of *speaker* classification and the local task of *phone* classification (distinct phonetic sounds that make up pronunciations of words). These two tasks are interesting for self-supervised representation learning as the former requires representations that discriminate speakers but are invariant to content, while the latter requires the opposite. Strong performance on both tasks thus suggests strong generalization and disentanglement.

**Experimental Details**  We follow the setup of Oord et al. [2018] unless specified otherwise and use a 100-hour subset of the publicly available LibriSpeech dataset [Panayotov et al., 2015]. It contains the utterances of 251 different speakers with aligned phone labels divided into 41 classes. These phone labels were provided by Oord et al. [2018] who obtained them by force-aligning phone sequences using the Kaldi toolkit [Povey et al., 2011] and pre-trained models on Librispeech [Panayotov, 2014]. We first train the self-supervised model consisting of five convolutional layers and one autoregressive module, a single-layer gated recurrent unit (GRU). After convergence, a linear multi-class classifier is trained on top of the context-aggregate representation $c^M$ without fine-tuning the representations. Remaining implementation details are presented in Appendix A.2.

**Results**  Following Table 3, we analyze the performance of models on phone and speaker classification accuracy. *Randomly initialized* features perform poorly, demonstrating that both tasks require complex representations. The traditional, hand-engineered *MFCC features* are commonly used in speech recognition systems [Ganchev et al., 2005], and improve over the random features, but provide limited linear separability on both tasks. On the speaker classification task, *CPC* and *GIM* outperform the *supervised* baselines despite their feature models having been trained without labels, and GIM without end-to-end backpropagation. In this setting, both *GIM* and *Greedy Supervised*, where individual layers are trained greedily with a supervised loss function, achieve similar results to their respective end-to-end trained counterparts (*CPC* and *Supervised*). When classifying phones, *CPC* does not reach the supervised performance ($64.9\%$ versus $77.7\%$). *GIM* achieves $62.5\%$, while *Greedy Supervised* accomplishes $73.4\%$. Thus, in contrast to the vision experiments (Section 4.1), we see similar differences in performance between the greedily trained models (*GIM* and *Greedy Supervised*) when compared to their respective end-to-end optimized counterparts (*CPC* and *Supervised*).

Overall, the discrepancy between better-than-supervised performance on the speaker task and less-than-optimal performance on the phone task suggests that GIM and CPC are biased towards extracting sequence-global features.

**Ablation study**  The local greedy training enabled by GIM provides a step towards biologically plausible optimization and improves memory efficiency. However, the autoregressive module $g_{ar}$ aggregates its inputs over multiple patches and employs Backpropagation Through Time (BPTT), which puts a damper on both benefits. In Table 4, we present results on the performance of ablated models that restrict the flow of gradients through time.

| Method | Accuracy (%) |
|---|---|
| **Speaker Classification** | |
| Greedy InfoMax (GIM) | 99.4 |
| GIM without BPTT | 99.2 |
| GIM without $g_{ar}$ | 99.1 |
| **Phone Classification** | |
| Greedy InfoMax (GIM) | 62.5 |
| GIM without BPTT | 55.5 |
| GIM without $g_{ar}$ | 50.8 |

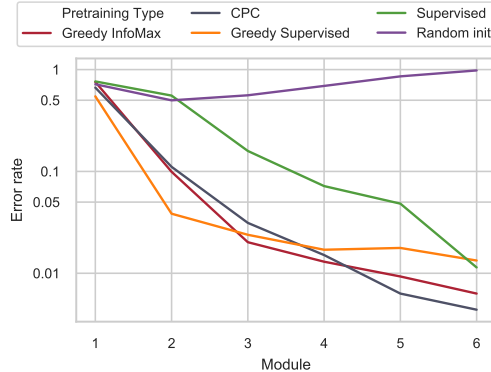

**Table 4:** Ablation studies on the LibriSpeech dataset for removing the biologically implausible and memory-heavy backpropagation through time.

**Figure 4:** Speaker Classification error rates on a log scale (lower is better) for intermediate representations (layers 1 to 5), as well as for the final representation created by the autoregressive layer (corresponding to the results in Table 3).

In order to limit the flow of gradients through time, we modify the autoregressive module. In general, the autoregressive module $g_{ar}$ takes the current input $z_t$, as well as the hidden state of the previous time-step $h_{t-1}$, in order to produce its output $c_t$, i.e. $c_t = g_{ar}(z_t, h_{t-1})$ (omitting the module-index $m$ here for brevity). In the standard GIM model, we block the flow of gradients to the previous module, such that $c_t = g_{ar}(\text{GradientBlock}(z_t), h_{t-1})$. In the ablation *GIM without BPTT*, we remove BPTT by blocking the flow of gradients between time-steps, such that $c_t = g_{ar}(\text{GradientBlock}(z_t), \text{GradientBlock}(h_{t-1}))$. For the ablation *GIM without $g_{ar}$*, we remove the autoregressive module entirely. Here, the linear classifier is applied to the representation created by the last encoding module (i.e. $z_t$).

In Table 4, we present the performance of the ablated models. Together, these two ablations indicate a crucial difference between the tested downstream tasks. For the phone classification task, we see a steady decline of the performance when we reduce the modeling of temporal dependencies, indicating their importance for solving this task. When classifying the speaker identity, reducing the modeling of temporal dependencies in the ablated models barely influences their performance.

Together with the image classification results from Section 4.1, where no autoregressive module was employed either, this indicates that the GIM approach performs best on downstream tasks where temporal or context dependencies do not need to be modeled by an autoregressive module. In these settings, GIM can outperform the CPC model, which makes use of end-to-end backpropagation, a global objective, and BPTT.

**Intermediate module representations** The greedy layer-wise training of GIM allows us to train arbitrarily deep models without ever running into a memory constraint. We investigate how the created representations develop in each individual module by training a linear classifier on top of each module and measuring their performance on the speaker classification task. With results presented in Figure 4, we first observe that each GIM module improves upon the representations of their predecessor. Interestingly, CPC exhibits similar performance in intermediate modules despite these modules relying solely on the error signal from the global loss function on the last module. This is in stark contrast with the supervised end-to-end model, whose intermediate layers lag behind their greedily trained counterparts. This suggests that, in contrast to the supervised loss, the InfoMax principle "stacks well", such that the greedy, iterative application of the InfoNCE loss performs similar to its global application.

## 5 Related Work

We have studied the effectiveness of the self-supervised CPC approach [Hénaff et al., 2019, Oord et al., 2018] when applied to gradient-isolated modules, freeing the method from end-to-end backpropagation. There are a number of optimization algorithms that eliminate the need for backpropagation altogether [Balduzzi et al., 2015, Kohan et al., 2018, Lee et al., 2015, Lillicrap et al., 2016, Ororbia et al., 2018, Scellier and Bengio, 2017, Xiao et al., 2019]. In contrast to our method, these

methods employ a global supervised loss function and focus on finding more biologically plausible ways to assign credit to neurons.

A recently published work by Nøkland and Eidnes [2019] likewise demonstrates that backpropagation-free layer-wise training is possible. Their similarity loss might be vaguely interpreted as another way of enforcing clustered representations. However, while our method achieves this entirely in a self-supervised fashion by clustering temporally or spatially nearby inputs, their similarity loss groups representations based on their class labels. Likewise, Belilovsky et al. [2019] showed that greedy layer-wise training with a supervised loss can scale to ImageNet. In an attempt to validate information bottleneck theory, Elad et al. [2018] develop a supervised, layer-wise training method that maximizes the mutual information between the outputs of a layer and the target whilst minimizing the mutual information between the inputs and outputs. In contrast to our proposal, these methods all rely on labeled data.

Jaderberg et al. [2017] develop decoupled neural interfaces, which enjoy the same asynchronous training benefits as Greedy InfoMax (GIM), but achieve this by taking an end-to-end supervised loss and locally predicting its gradients. Bengio et al. [2007], Hinton et al. [2006] focus on deep belief networks and propose a greedy layer-wise unsupervised pretraining method based on Restricted Boltzmann Machine principles, followed by optimizing globally using a supervised loss. Lee et al. [2009] use convolutional deep belief networks for unsupervised pretraining on the TIMIT audio dataset and then evaluate their performance by training supervised classifiers on top. Gao et al. [2018], Ver Steeg and Galstyan [2015] explore total correlation explanation, which is related to mutual information maximization, and show that it can be applied for layer-by-layer training.

Several recent works investigated the utilization of mutual information maximization in a representation learning setting [Belghazi et al., 2018, Hjelm et al., 2019, McAllester, 2018, Oord et al., 2018]. Poole et al. [2018] analyse these recent works under a common framework and highlight that InfoNCE exhibits low variance at a cost of high bias and propose new lower bounds that allow for balancing this bias/variance trade-off. However, the analysis of these improved bounds in the context of inter-patch mutual information optimization remains in order, and thus we focus on the original CPC InfoNCE loss to bias the learned representations towards slow features [Wiskott and Sejnowski, 2002].

Outside the information-theoretic framework, context prediction methods have been explored for unsupervised representation learning. A prominent approach in language processing is Word2Vec [Mikolov et al., 2013], in which a word is directly predicted given its context (continuous skip-gram). Likewise, Doersch et al. [2015] study such an approach for the visual domain. Similarly, graph neural networks use contrastive principles to learn unsupervised node embeddings based on their neighbors [Kipf and Welling, 2016, Nickel et al., 2011, 2015, Perozzi et al., 2014, Veličković et al., 2018]. Noise contrastive estimation has also been explored for independent component analysis [Hyvarinen and Morioka, 2016, 2017, Hyvarinen et al., 2018]. Schmidhuber [1992] proposes a method where individual features are minimized such that they cannot be predicted from other features, forcing them to extract independent factors that carry statistical information, at the risk of neurons latching onto local independent noise sources in the input.

## 6 Conclusion

We presented Greedy InfoMax, a novel self-supervised greedy learning approach. The relatively strong performance demonstrates that deep neural networks do not necessarily require end-to-end backpropagation of a supervised loss on perceptual tasks. Our proposal enables greedy self-supervised training, which makes the model less vulnerable to overfitting, reduces the vanishing gradient problem and enables memory-efficient asynchronous distributed training. While the biological plausibility of our proposal is limited by the use of negative samples and within-module backpropagation, the results provide evidence that the theorized self-organization in biological perceptual networks is at least feasible and effective in artificial networks, providing food for thought on the credit assignment discussion in perceptual networks [Bengio et al., 2015, Linsker, 1988].

**Acknowledgments**

We thank Jorn Peters, Marco Federici, Rudy Corona, Pascal Esser, Joop Pascha and the anonymous reviewers for their insightful comments. This research was supported by Philips Research and the NVIDIA GPU Grant.

## Footnotes

[1]Our code is available at https://github.com/loeweX/Greedy_InfoMax.

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
