[Supplementary Material · Appendix_Greedy_InfoMax.pdf]

## A    Experimental Setup

We use PyTorch [Paszke et al., 2017] for all our experiments.

### A.1    Vision Experiments

In our vision experiments, we employ the ResNet-50 v2 architecture [He et al., 2016b], in which we remove the max-pooling layer and adjuste the first convolutional layer in such a way that the size of the feature map stays constant. Thus, the first convolutional layer uses a kernel size of 5, a stride of 1 and a padding of 2. Additionally, we do not employ batch normalization [Ioffe and Szegedy, 2015].

We train our model on 8 GPUs (GeForce 1080 Ti) each with a minibatch of 16 images. We train it for 300 epochs using Adam and a learning rate of 1.5e-4 and use the same random seed in all our experiments.

For the self-supervised training using the InfoNCE objective, we need to contrast the predictions of the model for its future representations against negative samples. We draw these samples uniformly at random from across the input batch that is being evaluated. Thus, the negative samples can contain samples from the same image at different patch locations, as well as from different images. We found that including the positive sample (i.e. the future representation that is currently to be predicted) in the negative samples did not have a negative effect on the final performance. For each evaluation of the InfoNCE loss, we use 16 negative samples and predict up to $k = 5$ rows into the future. For contrasting patches against one another, we spatially mean-pool the representations of each patch.

Before applying the linear logistic regression classifier on the output of the third residual block, we spatially mean-pool the created representations of size $7 \times 7 \times 1024$ again. Thus, the final representation from which we learn to predict class labels is a 1024-dimensional vector. We use the Adam optimizer for the training of the linear logistic regression classifier and set its learning rate to 1e-3. We optimized this hyperparameter by splitting the labelled training set provided by the STL-10 dataset into a validation set consisting of $20\%$ of the images and a corresponding training set with the remaining images.

### A.2    Audio Experiments

The detailed description of our employed architecture is given in Table 5. We train our model on 4 GPUs (GeForce 1080 Ti) each with a minibatch of 8 examples. Our model is optimized with Adam [Kingma and Ba, 2014] and a learning rate of 2e-4 for 300 epochs. We use the same random seed for all our experiments. Overall, our hyperparameters were chosen to be consistent with Oord et al. [2018].

**Table 5:** General outline of our architecture for the audio experiments.

| Layer | Output Size | Parameters | | |
|---|---|---|---|---|
| | (Sequence Length $\times$ Channels) | Kernel | Stride | Padding |
| Input | $20480 \times 1$ | | | |
| Conv1 | $4095^a \times 512$ | 10 | 5 | 2 |
| Conv2 | $1023^a \times 512$ | 8 | 4 | 2 |
| Conv3 | $512^a \times 512$ | 4 | 2 | 2 |
| Conv4 | $257^a \times 512$ | 4 | 2 | 2 |
| Conv5 | $128 \times 512$ | 1 | 2 | 1 |
| GRU | $128 \times 256$ | - | - | - |

[a]For applying the InfoNCE objective on these layers, we randomly sample a time-window of size 128 to decrease the dimensionality.

Similarly to the vision experiments, we take the negative samples uniformly at random from across the batch that is currently evaluated. Again this may include the positive sample. In our audio experiments, we use a total of 10 negative samples and predict up to $k = 12$ time-steps into the future.

We train the linear logistic regression classifier using the representations of the top, autoregressive module without pooling. Again, we employ the Adam optimizer but select different learning rates than before. For this hyperparameter search, we split the training set provided by Oord et al. [2018] into two

random subsets using $25\%$ of the samples as a validation set. In the speaker classification experiment, we used a learning rate of 1e-3, while we set it to 1e-4 for the phone classification experiment.