[Reviews · NeurIPS 2019]

Reviewer 1



In the present manuscript the authors propose greedy InfoMax, a greedy algorithm which allows unsupervised learning in deep neural networks with state of the art performance. Specifically, the algorithm leverages implicit label information which is encoded temporally in the streaming data. Importantly, the present work rests on the shoulders and success of Contrastive Predictive Coding, but dispenses with end-to-end training entirely. Getting greedy layer-wise unsupervised learning to perform at such levels is quite impressive and will without doubt have an important impact on the community. The work is original and the quality of the writing and figures seems quite high. What I would have liked to see is a more in depth review of the precise data generation process. To understand approximately what was done, I had to go back two generations of papers. It would be useful to make the manuscript a little more self-contained in that regard. When comparing to the results from Nokland and Eidnes it might be worth noting that their similarity loss (although computed from labels) does not fully use the label information, but merely uses the information of which samples belong to the same class and which ones belong to different classes. I feel this is similar in spirit to the present approach where this grouping is done in time. It might be worth mentioning this where Nokland and Eidnes is discussed. The introduction paragraph on the implausibility of backprop could be moderated a little. It seems to be one of the open questions at the moment whether and how the brain backpropagates information. The stronger and less controversial point is rather the inaccessibility of label information. Finally, this might not be feasible for computational reasons, but it would be nice to report error margins on the accuracy results in Table 1 and 2 for different initial conditions of the network. UPDATE: Thanks for the detailed answers to the reviewers. I think this is an important contribution. I am looking forward to the error bars and the improved description of the (non-trivial) data generation/preprocessing steps. Also, any insights and discussion on the underlying connections to the similarly matching loss case of Nokland et al. would be welcomed.

Reviewer 2



Quality: I view this as a significant result. It is interesting that a competitive self-supervised learning scheme exists which only relies on limited backpropagation, and this understanding might be a step towards biologically plausible learning rules. Clarity: I found this paper and methods to be reasonably clear, but I had several questions about the techniques. In particular, I don’t understand how the authors implemented an autoregressive model without backpropagation through time Originality and Significance: This work seems to be original and significant. Neural systems receive large amounts of unlabeled data, and need to draw semantic features from them a biologically plausible way, and this might be a step towards understanding how this is accomplished. In addition, these schemes might have benefits for distributed training.

Reviewer 3



Authors proposed a biologically plausible self supervised learning method. They utilized previously published work and information theory. They also support their ideas with experiments.

[Author Response · NeurIPS 2019]

First of all, we would like to thank you all for your time and thoughtful comments on our manuscript.

Since our submission to NeurIPS, we continued to develop our method and managed to further improve our results. Initially, we were suspicious whether greedy layer-wise training could indeed match end-to-end trained models in performance, but conducting our experiments repeatedly yields consistent performance. We are now in the process of extracting confidence bounds and releasing our code base in order to allow the community to scrutinize our findings.

**Reviewer 1** - Thank you very much for your review and the positive feedback on our method.

We appreciate your feedback to make the manuscript more self-contained and to include a more in-depth review of the precise data generation process. We will incorporate this by providing more details on the dataset that we used in our audio experiments, more specifically the phone labels that are not part of the original Librispeech dataset. These were provided by Oord et al. (2018) who obtained them by force-aligning phone sequences using the Kaldi toolkit (Povey et al., 2011) and pre-trained models on Librispeech (Panayotov, 2014). We will add this clarification in our final manuscript.

Your observation that the similarity loss of Nøkland and Eidnes (2019) has similarities to InfoNCE is very interesting and might path the way for future research on layer-wise training. As such we will include this in our discussion of their work.

There are certainly more points to discuss on whether and how the brain backpropagates information. We are happy to use the additional space of the final manuscript to provide a more in-depth discussion on this topic, including more recent theories on how neural circuits in the brain could approximate the error back-propagation algorithm (Whittington and Bogacz, 2019).

We agree that including error margins on our accuracy results can validate the stability of the training and significance of our results. We are actively working to add them to our manuscript.

**Reviewer 2** - Thank you very much for your review.

We agree that the experimental setup of the ablation studies could be clarified. In the following, we provide a more thorough description which we will also incorporate in our final manuscript:

In the forward pass, the output $c_t$ for time-step $t$ of the autoregressive module $g_{ar}$ is generated by taking into account the hidden state of the previous time-step $h_{t-1}$, as well as the current input $z_t$, i.e. $c_t = g_{ar}(z_t, h_{t-1})$ (omitting the module-index $m$ here for brevity). For the backward pass in the standard GIM model, we block the flow of gradients to the previous module. We can express this using the gradient blocking operator as defined in the draft ($\text{GradientBlock}(x) \triangleq x, \nabla \text{GradientBlock}(x) \triangleq 0$), such that $c_t = g_{ar}(\text{GradientBlock}(z_t), h_{t-1})$. In the ablation study in which we remove backpropagation through time ("GIM without BPTT"), we additionally block the flow of gradients between time-steps, such that the gradients derived from the loss at time-step $t$ do not influence the calculation of the hidden state of the previous time-step $h_{t-1}$. Thus, $c_t = g_{ar}(\text{GradientBlock}(z_t), \text{GradientBlock}(h_{t-1}))$. In both of these models, we train the linear classifier on top of the representation $c_t$ for the downstream tasks. When we remove the autoregressive module entirely ("GIM without $g_{ar}$"), the linear classifier is applied on the representation created by the last convolutional module (i.e. $z_t$).

**Reviewer 3** - Thank you for your feedback.

Since no points for improvements were brought up, we focused our discussion on the points raised by reviewers 1 and 2 instead.

# References

A. v. d. Oord, Y. Li, and O. Vinyals, "Representation learning with contrastive predictive coding," *arXiv preprint arXiv:1807.03748*, 2018.

D. Povey, A. Ghoshal, G. Boulianne, L. Burget, O. Glembek, N. Goel, M. Hannemann, P. Motlicek, Y. Qian, P. Schwarz *et al.*, "The kaldi speech recognition toolkit," in *IEEE 2011 workshop on automatic speech recognition and understanding*, no. CONF. IEEE Signal Processing Society, 2011.

V. Panayotov, "Kaldi pretrained model on LibriSpeech SAT and DNN," http://www.kaldi-asr.org/downloads/build/6/trunk/egs/librispeech/, 2014, [Online; accessed 29-July-2019].

A. Nøkland and L. H. Eidnes, "Training neural networks with local error signals," in *Proceedings of the 36th International Conference on Machine Learning*, 2019.

J. C. Whittington and R. Bogacz, "Theories of error back-propagation in the brain," *Trends in cognitive sciences*, 2019.


[Meta-Review · NeurIPS 2019]

After discussion, the reviewers agree that the paper proposes important work and that the results are state of the art for several audio and image processing tasks.